# Mexican Native Black Bean Anthocyanin-Rich Extracts Modulate Biological Markers Associated with Inflammation

**DOI:** 10.3390/ph16060874

**Published:** 2023-06-13

**Authors:** Jonhatan Contreras, Montserrat Alcázar-Valle, Eugenia Lugo-Cervantes, Diego A. Luna-Vital, Luis Mojica

**Affiliations:** 1Food Technology, Centro de Investigación y Asistencia en Tecnología y Diseño del Estado de Jalisco, Unidad Zapopan, Camino Arenero 1227, El Bajío del Arenal, Zapopan 45019, Jalisco, Mexico; jocontreras_al@ciatej.edu.mx (J.C.); ealcazar@ciatej.mx (M.A.-V.); elugo@ciatej.mx (E.L.-C.); 2Tecnologico de Monterrey, The Institute for Obesity Research, Avenida Eugenio Garza Sada 2501, Monterrey 64849, NL, Mexico; dieluna@tec.mx

**Keywords:** native black bean, phenolic compounds, anthocyanins, countercurrent chromatography, antioxidant potential, anti-inflammatory potential

## Abstract

This work aimed to obtain and characterize anthocyanin-rich extracts (ARE) from native black beans and evaluate their antioxidant and anti-inflammatory potential. The initial extract was obtained by supercritical fluids (RE) and purified using Amberlite^®^ XAD-7 resin (PE). RE and PE were fractionated using countercurrent chromatography, and four fractions were obtained (REF1 and REF2 from RE, PEF1, and PEF2 from PE). ARE and fractions were characterized, and the biological potential was evaluated. ABTS IC_50_ values ranged from 7.9 to 139.2 (mg C3GE/L), DPPH IC_50_ ranged from 9.2 to 117.2 (mg C3GE/L), and NO IC_50_ ranged from 0.6 to143.8 (mg C3GE/L) (*p* < 0.05). COX-1 IC_50_ ranged from 0.1 to 0.9 (mg C3GE/L), COX-2 IC_50_ ranged from 0.01 to 0.7 (mg C3GE/L), and iNOS IC_50_ ranged from 0.9 to 5.6 (mg C3GE/L) (*p* < 0.05). The theoretical binding energy for phenolic compounds ranged from −8.45 to −1.4 kcal/mol for COX-1, from −8.5 to −1.8 kcal/mol for COX-2, and from −7.2 to −1.6 kcal/mol for iNOS. RE and REF2 presented the highest antioxidant and anti-inflammatory potential. Countercurrent chromatography effectively isolates and purifies bioactive compounds while maintaining their biological potential. Native black beans present an attractive phytochemical profile and could be used as ingredients in nutraceuticals and functional foods.

## 1. Introduction

Common beans (*Phaseolus vulgaris* L.) are native legumes from Mexico; moreover, they are an important component of the traditional Mexican diet [1,2]. Common bean’s cotyledon nutritional components include polysaccharides, proteins, fiber, and a small concentration of lipids. Their coats contain a significant amount of fiber and different phytochemicals such as saponins, tannins, phenolic compounds, and phytosterols [3,4,5].

There are several reports related to the diversity of Mexican native common beans, with Chiapas having a great diversity and an important number of varieties such as white, purple, red, yellow, brown, and black bean [6,7]. Recent studies have evaluated the biological potential, physicochemical, and nutritional properties of native beans from the *Phaseolus* genera, including *P. vulgaris*, *P. lunatus*, *P. polyanthus*, and *P. coccineous* species. These beans were collected in Mexico’s South Pacific Region, including Guerrero, Oaxaca, and Chiapas states. Local farmers select and preserve the best seeds for generations. This seed selection process could influence the phytochemical and nutritional content. Moreover, those beans are a part of the culture and identity of the people in that region [8,9,10]. Black bean is the most produced and consumed legume in Mexico, and several studies have reported their nutritional and phytochemical content, including anthocyanins, responsible for the black color [5,11,12,13].

Alcázar-Valle et al. [8] described the physicochemical characteristics, phytochemical profile, and antioxidant potential of a large collection of common beans from Chiapas. They found an interesting black bean variety called “negro de vaina morada”, which presented a high total anthocyanin content.

Anthocyanins are a natural group of red–purple and blue pigments [14,15,16]. This group of molecules contains three aromatic rings with hydroxyl groups that are substituted to saturated or glycosylated chains, making their structures very reactive and highly unstable under different conditions such as pH, solvents, and temperature, among others [14,16]. Anthocyanins are important for the food industry because of their capacity to be used as natural food colorants and their biological potential [15,17]. Therefore, several studies of anthocyanin-rich extracts (ARE) from different sources have demonstrated their capacity to modulate markers related to oxidative stress, obesity, hypertension, type-2 diabetes, heart attack, and some types of cancer [18,19]. Figure 1 shows the structure of the three main anthocyanins found and reported in black beans [10,20,21,22]. The figure shows that the three molecules are different because they have different substituents, including methoxy and hydroxy groups in R_1_ and R_2_ positions in the phenolic ring.

Inflammation is the biological process where cells respond to their environment when they detect a physicochemical change or mechanical stress [23,24]. Inflammation could increase oxidative stress in cells and tissues by producing reactive oxygen species (ROS) [25,26]. Black bean anthocyanin’s anti-inflammatory and anti-diabetes biological potential has been reported [21,22,27].

Isolating and stabilizing anthocyanins from natural sources are important challenges for their successful application as food ingredients [18,28]. There are several extraction and purification methods; however, extraction and purification technologies commonly used present low selectivity, low yields, and poor stability of the compound of interest [29,30]. Therefore, it is necessary to use methodologies to extract and purify compounds efficiently and selectively [31]. For instance, supercritical fluid extraction (SFE) is a modern technique used to extract phytochemicals from a plant matrix using supercritical CO_2_ as the main solvent. However, CO_2_ is a non-polar molecule, making this compound an unsuitable solvent for extracting polar compounds. Therefore, other polar molecules such as ethanol, methanol, and water are used as co-solvents during the process. Increasing the solubility of the solvent system and producing an increment in the extraction yield for polar bioactive compounds [32,33]. Another alternative for purifying and fractionating compounds in mixtures is high-pressure countercurrent chromatography (HPCCC) [34,35]. The HPCCC is a type of chromatography that uses two liquid phases of different solvents with gravitational and centrifugal forces, separating the compounds of interest [36,37].

HPCCC has been used to obtain anthocyanin-rich fractions from different food matrices. Winterhalter et al. [38,39,40] reported the extraction and fractionation of anthocyanins from natural sources such as *Impatiens glandulifera*, *Vaccinium vitis-idaea* L., black chokeberry, cranberry, and pomegranate (*Punica granatum* L.) using HPCCC. The fractions obtained presented high anthocyanin concentration and a purified profile of the molecules of interest. HPCCC permits the obtaining of concentrated anthocyanin fractions with unique compositions from natural sources using less solvent and time [34,37,39,40]. However, few studies reported the biological potential of the generated fractions, physicochemical characteristics, and their phytochemical content. In this sense, HPCCC anthocyanin purified fractions could be characterized, and their biological potential and stability as colorants could be evaluated.

This work aimed to obtain and characterize anthocyanin-rich extracts and fractions from Chiapas native black beans obtained using HPCCC and column purification and evaluate their antioxidant and anti-inflammatory potential using biochemical assays.

## 2. Results and Discussion

### 2.1. Study Description

A raw extract (RE) was obtained from native black beans by supercritical fluids technology. Then, RE was purified with Amberlite^®^ XAD-7HP resin in a column (PE). RE and PE were injected into the HPCCC equipment to obtain anthocyanin-rich fractions. Two fractions from RE were collected accordingly with the anthocyanin retention time and absorption in the detector (REF1 and REF2). Similarly, another two fractions from PE were obtained (PEF1 and PEF2). Treatments (extracts (RE, PE) and fractions (REF1, REF2, PEF1, PEF2)) were evaluated to determine and identify the total phenolic and anthocyanin content. Some assays were evaluated in all treatments to probe the biological potential, including antioxidant and anti-inflammatory potential. Finally, an in silico assay was performed to predict the theoretical interaction of the identified molecules of the common bean extract and fractions and molecular markers related to inflammation.

### 2.2. Phenolic and Anthocyanin Quantification

Table 1 presents black bean extracts’ total phenolic and anthocyanin content. The highest content of TPC was 358.4 mg GAE/L for RE and 315.0 mg GAE/L for REF2. After HPCCC fractionation, REF2 presented the highest TPC concentration compared to other fractions. This could be due to the higher initial concentration of these compounds on RE. On the other hand, Amberlite^®^ resin forms bonds or intermolecular interactions between the compounds, which complicate their elution with the solvent, decreasing their concentration in the final PE. As a result, treatments that interacted with Amberlite^®^ (PF, PEF1, and PEF2) presented lower TPC compared to the non-purified fractions (RE, REF1, and REF2) (*p* ˂ 0.05).

TPC of Chiapas native black bean (RE) was 2- to 6-fold higher (50.1 mg GAE/g dry coat (DC)) compared to other reports that used Black, Pinto, and June flower common bean Mexican cultivars whose concentrations ranged from 6.0 to 17.3 mg GAE/g DC [20,21,27] using liquid-solid extraction. Hsieh-Lo et al. [41] used supercritical fluids to extract the phenolic compounds from the “Negro San Luis” black commercial variety and reported 11.1 mg GAE/g DC, which is five times lower compared to the concentration reported in this work. Results show that the studied native variety contains higher concentrations of phenolic compounds compared to commercial and noncommercial Mexican black varieties. The phytochemical concentration could be affected by several factors, such as the cultivar type, extraction method, solvents used, the harvesting year, grown conditions, and analytic method, among others [2,3,28].

Regarding the total anthocyanins content (TAC), REF2 treatment presented the highest concentration, 66.3 mg C3GE/L, followed by RE (52.2 mg C3GE/L) (*p* < 0.05). PE and REF1 presented similar concentrations (28.1 mg C3GE/L and 27.8 mg C3GE/L, respectively (*p* > 0.05)). The lowest concentrations of TAC were for PEF2 (13.1 mg C3GE/L) and PEF1 (10.7 mg C3GE/L). The processes to obtain the different extracts (Amberlite^®^ or HPCCC) efficiently increased the TAC. The fractionation using HPCCC from the crude extract retained and concentrated most of the anthocyanins present in the RE. On the other hand, the fractions obtained from the purified extract contained lower concentrations compared to the crude extract-derived fractions.

Mojica et al. [20,27] and Aguilera et al. [21] reported extracts of different Mexican commercial and native black bean varieties with concentrations ranging from 1.0 to 2.5 mg C3GE/g DC for TAC. Results presented in this work (8.0 mg C3GE/g DC) are higher in TAC compared to those reports. The differences in concentration of anthocyanins could be due to the location and conditions where seeds were cultivated, the weather, and other conditions, including the extraction methodology [42]. Hsieh-lo et al. [41] reported a lower TAC in an extract of the black bean variety “San Luis” obtained using supercritical fluids (2.6 mg C3GE/g DC), Alcázar-Valle et al. [8] also reported a high anthocyanin content in the “negro de vaina morada” from Chiapas black bean cultivar.

TPC and TAC results showed that the more concentrated fractions were RE and REF2. RE before any treatment showed higher phenolic compounds concentration compared to PE, indicating that Amberlite^®^ resin interacts with phenolic compounds [43]. Consequently, REF2 contained the majority of compounds that were originally in RE. Additionally, REF2 was obtained using HPCCC directly, which influenced the molecule profile separation [39,40]. This technique efficiently isolates the molecules of interest compared to Amberlite^®^ resin. The resin purifying step could be removed, and HPCCC could be used to purify and fractionate raw extracts.

### 2.3. Tentative Identification and Quantification of Phenolic Compounds in Treatments

Regarding the HPCCC method, the four solvents in the liquid mixture (water, hexane, ethyl acetate, and methanol) formed the upper and lower phases. The upper liquid phase was used as a stationary phase, and then the lower phase was used as a mobile phase through the column to separate the anthocyanin and phenolic compounds. The upper phase was a non-polar liquid mixture, while the lower phase was a polar liquid mixture. Phenolic compounds and anthocyanin are molecules with a polar structure, showing more affinity to the lower phase [12,13].

Accordingly, the first compounds eluted were those with high polarity present in the extracts (RE and PE), such as sugars and structures derived from these (0–24 min). Then, phenolic compounds and anthocyanins with intermediate polarity were eluted in a mixture (24–27 min) [12,16]. Other uncolored phenolic compounds, including flavonoids and phenolic acids present in the extracts, have a higher structural polarity compared to the anthocyanins and were collected in the first fractions of both extracts [34,39]. REF1 and PEF1 mainly contain uncolored phenolic compounds, while REF2 and PEF2 contain many colored components, principally glycosylated anthocyanins. At the end of the method (27–35 min), those components with a more non-polar structure were not collected.

Table 2 shows the identified non-colored and colored phenolic compounds by direct analysis in real-time mass spectrometry (DART-MS). Seventeen phenolic compounds were tentatively identified, including p-coumaric acid, ferulic acid, caffeic acid, rosmarinic acid, gallic acid, protocatechuic acid, glycitin, naringenin, genistein, rutin, catechin, myricetin, quercetin-3-galactoside, delphinidin-3-glucoside, cyanidin-3-glucoside, petunidin-3-glucoside, and malvidin-3-glucoside. Previous studies using different black bean varieties have identified these chemical compounds [20,21,27].

The phytochemicals identified were different among treatments. RE and PE showed 17 phenolic compounds (Table 2). Afterward, PEF1 did not contain delphinidin-3-glucoside, and PEF2 did not contain rutin and catechin. Moreover, rutin, catechin, myricetin, and p-coumaric acid were not identified in REF1. Delphinidin-3-glucoside, genistein, rutin, naringenin, catechin, myricetin, and rosmarinic acid were not found in REF2. Fractioning and purifying processes modified the phytochemical profile in each treatment as a consequence of the affinity of the molecules for the resin and the solvent phases in the HPCCC process [44]. Raw extract fractions (REF1, REF2) presented the lowest number of compounds compared to purified extract fractions (PEF1, PEF2), which showed most of the compounds contained in the original extract. These results evidenced that resin purification decreased the concentration of phenolic compounds and removed other compounds, such as glucosylated derivatives and sugars, that could interfere during mass analysis. Moreover, cleaner PE fractions facilitate the identification process [43]. In this sense, PE fractions show compounds not identified in RE fractions.

Table 3 shows the quantification results of some phenolic compounds and anthocyanins detected in the different extracts and fractions by high-pressure liquid chromatography (HPLC). Phenolic compounds were organized as phenolic acids, flavonoids, and glucosylated components. Regarding the phenolic acids, gallic acid was quantified in the fractions of RE and REF2 (180.1 and 170.4 mg/L). For flavonoids, it was possible to quantify genistein REF2 (35.2 mg/L) and myricetin in PE (10.4 mg/L). On the other hand, the glucosylated phenolic compounds in the samples were the most present group quantified in the fractions. Apigenin-7-glucoside was found in REF2 (15.2 mg/L), epigallocatechin gallate in RE, REF2, and PE (from 5.3 to 14.8 mg/L). Likewise, kaempferol-3-glucoside was quantified in RE and PE and fractions REF2 and PEF2 (from 25.3 to 39.5 mg/L). Quercetin-3-glucoside was the compound quantified in all samples from 10.2 to 70.4 (PE) mg/L, and quercetin-3-galactoside was found in REF1, PEF1, and PE treatments (from 3.9 to 15.2 mg/L). Appendix A shows the chromatograms obtained by HPLC analysis of the extracts and fractions. Phenolic compounds were quantified at 280 nm (Appendix A), and Anthocyanins were quantified at 520 nm (Appendix A).

Results showed that the extracts present a high abundance of glucosylated compounds. Moreover, the fractionation profile shows that quercetin-3-glucoside (Q3G) was present in all fractions, indicating its stability and solubility at different conditions. Similarly, kaempferol-3-glucoside was identified in the second fractions of the extracts (REF2 and PEF2), indicating that this compound interacted better with the stationary phase in HPCCC for both RE and PE [44]. At the same elution time in the HPCCC, quercetin-3-galactoside was present in PE, REF1, and PEF1.

López et al. [45], Moreno-Jiménez et al. [46], and Duenas et al. [47] reported the presence of several phenolic compounds in common bean, including hesperetin-7-rutinoside and derivates, naringenin and naringenin glucosides, quercetin-3-O-galactoside, quercetin-3-O-glucoside, kaempferol-3-O-glucoside, kaempferol, myricetin-3-O-glucoside, apigenin-7-O-glucoside, daidzein derivatives, genistein derivatives, delphinidin-3-O-glucoside, cyanidin-3-O-glucoside, pelargonidin-3,5-diglucoside, pelargonidin-3-O-glucoside, petunidin derivatives, malvidin derivatives, gallic acid, protocatechuic acid, ferulic acid, trans-p-coumaric acid, sinapyl aldaric acid, and sinapic acid.

The three most reported anthocyanins in black bean [20] were quantified (Table 3). Delphinidin-3-glucoside was quantified in all the treatments (from 0.71 to 40.33 mg/L); moreover, REF2 showed the highest content, followed by RE. Malvinidin-3-glucoside was not quantified in REF1; however, it was present in high amounts in both RE, REF2, and PEF2 fractions (from 5.91 to 29.00 mg/L). Petunidin-3-glucoside was presented in high concentration in REF2 and RE, followed by PEF2, REF1, and PE, while PEF1 was the treatment with the lowest concentration (from 1.65 to 35.84 mg/L). Other reports of black bean anthocyanin profiles included other anthocyanins that were not identified or quantified in this work, such as malvidin-3,5-diglucoside, malvidin-3-O-galactoside, pelargonidin-3,5-O-diglucoside, cyanidin-3-O-glucoside, pelargonidin-3-O-glucoside, cyanidin-3-O-(6″-malonyl) glucoside, pelargonidin-3-O-(6″-malonyl) glucoside [46,48,49].

Hsieh-Lo et al. [41] reported catechin, chlorogenic acid, ferulic acid, gallic acid, syringic acid, myricetin-3-O-glucoside, p-coumaric acid, rutin hydrate, myricetin, quercetin, and kaempferol compounds. Those compounds were extracted using supercritical fluids from a commercial variety of Mexican black beans. In contrast, Fonseca et al. [10] reported quercetin-3-D-galactoside, cyanidin-3-glucoside, gallic acid, caffeic acid, daidzin, sinapic acid, naringenin, rosmarinic acid, catechin, myricetin, and ferulic acid in raw and purified native black bean extracts. Native black bean phytochemical profile varies according to the cultivar, seed selection, and farming conditions where they were cultivated [50].

The results showed that HPCCC is highly effective in preserving and concentrating anthocyanins in the extract. REF2 and PE contained most phenolic compounds and anthocyanins identified in RE. HSCCC (high-speed countercurrent chromatography) is similar to HPCCC; the difference is that HPCCC increases the g-force and the column length producing a ten-times increase in yield thanks to the better mobile phase flow rates and higher stationary phase retention [51]. Degenhardt et al. [52] noted that HSCCC provides an important contribution to the isolation of pure standards because it was possible to obtain a high purification level of some red wine anthocyanins. They also mentioned that the process opened the possibility of obtaining anthocyanin standards that are not commercially available [52]. This work showed that it was possible to purify and enrich the phenolic compounds and anthocyanins in the raw extract to one of the fractions (REF2) without using another purifying process. Du et al. [53] reported that HSCCC was an effective technique for the selective preparation and purification of anthocyanins at the laboratory scale. They purified delphinidin- and cyanidin-3-O-sambubiosides from bilberry (*Vaccinium myrtillus* L.) without using solid-phase column chromatography.

Countercurrent chromatography is an efficient alternative for the separation and purification of natural bioactive compounds. The fraction REF2 obtained by HPCCC presented the highest phenolic compounds and anthocyanins content, similar to the RE. The efficiency of HPCCC was compared with the column purification process. HPCCC is a good methodology to isolate and purify bioactive compounds.

### 2.4. Biological Potential

The antioxidant potential of treatments was related to the ability of the phenolic compounds and anthocyanins present in the extracts and fractions to neutralize oxygen free radicals that were generated in excess directly as the 2,2′-azino-bis(3-ethylbenzothiazoline-6-sulfonic acid (ABTS) and the 2,2-diphenyl-1-picrylhydrazyl (DPPH) or indirectly as the nitric oxide (NO) to the reaction environment. This antioxidant capacity could be related to activating the body’s defense mechanisms that prevent the effects of oxidative stress [14,41,54].

The results obtained for antioxidant potential are presented in Figure 2. ABTS results (Figure 2a) show a similar radical inhibition for RE, REF2, PE, and PEF2, ranging from 0.5 to 4 mg C3GE/L (*p* > 0.05). REF1 and PEF1 presented lower antioxidant potential ranging from 135.0 to 150.0 mg C3GE/L (*p* < 0.05), compared to RE, REF2, PE, and PEF2.

DPPH results (Figure 2b) show a significant difference among treatments REF1, PEF1, and PEF2 with low antioxidant potential (from 60.0 to 120.0 mg C3GE/L for IC_50_) and high values to RE, PE, and REF2 (from 5.0 to 20.0 mg C3GE/L for IC_50_) which demonstrates a positive correlation among radical inhibition and high concentration of phenolic compounds and anthocyanins. It is important to mention that PE and RE extracts and the REF2 fraction obtained by HPCCC presented similar antioxidant potential.

The NO assay is based on NO production from the sodium nitroprusside simulating physiological conditions, which interacts with oxygen in the medium and produces nitrite ions. These last species react with Griess reagent development of a chromophore to detect and quantify. As NO is considered a free radical because of its unpaired electron, this assay measures the antioxidant potential of compounds because the inhibitors compete for oxygen and decrease nitrite ion production [54,55].

PEF2 had the lowest potential (147.9 mg C3GE/L), then PEF1 exhibited 43.3 mg C3GE/L. The highest potential was for RE, REF1, REF2, and PE, which exhibit an IC_50_ ranging from 0.3 to 0.8 mg C3GE/L (*p* > 0.05). These results (Figure 2c) show that compounds in RE and PE have similar antioxidant potential compared to REF1 and REF2.

Applied antioxidant methodologies give evidence that raw extracts and fractions show the capacity to inhibit different free radicals. Therefore, this antioxidant potential could be associated with black bean phenolic compounds and anthocyanins [28,42,49]. The antioxidant potential of compounds from foods is relevant due to their potential to reduce cellular oxidative stress, which is related to the development of numerous health problems and illnesses [15,56,57].

Fonseca et al. [10] reported IC_50_ of 0.2 mg/mL for purified and 2.3 mg/mL raw black bean extracts for ABTS and IC_50_ of 0.05 mg/mL for raw black bean extract for DPPH. Ombra et al. [58] reported that a common bean water extract exhibited an IC_50_ of 1.5–55.2 mg/mL for DPPH assay. Romani et al. [59] reported that a Sarconi bean 70% ethanol extract IC_50_ ranged from 2.78 to 16.9 (mg/L) for DPPH assay. These results are higher compared to the obtained in this work. The difference could be related to the extraction method and ethanol concentration, influencing the final bioactive compounds concentration [60].

Mojica et al. [20] reported inhibition percentages ranging from 5.7 to 85.4 for the NO assay using phenolic compounds extracts (20 g/L) from Pinto, Flor de Mayo, and Flor de Junio, Black, and Carioca common bean varieties. According to the results, RE, REF2, and PE have a similar antioxidant potential (*p* ˃ 0.05) which could be related to the TAC in these treatments. Other works [22,61,62] have reported a positive correlation between the concentration of anthocyanins and the antioxidant potential. Anthocyanins are highly reactive molecules that interact with free radicals and inhibit their action [16,31,62]. Therefore, despite the loss of phenolic compounds and anthocyanins concentration in the Amberlite^®^ resin, the recovered compounds maintain their antioxidant potential, as well as those that were directly purified by HPCCC.

Inflammation is the biological process where cells respond to their environment when they detect a physicochemical change or mechanical stress. This condition, in general, is presented before other physiological pathologies such as obesity, hypertension, and type 2 diabetes [63]. It is known that molecules such as phenolic compounds present in natural products have the capacity to prevent and modulate inflammatory markers [64]. Some studies have reported that black beans anthocyanin could modulate molecular markers related to these diseases [22,65]. Nevertheless, there are no reports that evaluate how the biological potential is retained or modified after the extracts pass for purifying and concentrating process.

The anti-inflammatory potential of the extracts and fractions (Figure 3) was evaluated by inhibiting enzymes related to the inflammation process, such as inducible nitric oxide synthase (iNOS), cyclooxygenase 1 (COX-1) and cyclooxygenase 2 (COX-2).

iNOS, COX-1, and COX-2 produce mediators (nitric oxide or prostaglandins) to cause inflammation and tissue damage [64]. iNOS produces cellular nitric oxide in tissues. This enzyme is specialized in producing this molecule in response to alterations and lack of oxygen; therefore, its activation produces the oxide that triggers an inflammatory response and the secretion of cytokines, and the recruitment of macrophages to the tissue [56,66,67]. Cyclooxygenases (COX-1 and COX-2) are enzymes that catalyze the production of prostaglandins when stimulated in the cellular signaling cascades. These molecules promote cytokine synthesis, which contributes to the pro-inflammatory stage of the tissue [56,68,69].

iNOS exhibited a low inhibition (Figure 3a) for PEF1 (5.02 mg C3GE/L), followed by PEF2 (4.32 mg C3GE/L. REF1 presented 3.34 mg C3GE/L potential, then REF2 presented 1.26 mg C3GE/L potential. The highest inhibition was obtained for RE and PE (1.12 and 1.09 mg C3GE/L) with no significant differences (*p* ˃ 0.05).

COX-1 inhibition results are presented in IC_50_ values (Figure 3b). The highest potential was presented by RE and PEF1 ranging from 0.17 to 0.21 mg C3GE/L. Then, RFE2 presented 0.25 mg C3GE/L. PE and REF1 had similar values ranging from 0.78 to 0.82 mg C3GE/L. The lowest potential was found for PEF2, which showed a 0.92 mg C3GE/L potential (*p* ˂ 0.05).

COX-2 showed higher inhibition (Figure 3c) by PE (0.04 mg C3GE/L) than REF1 and REF2 (0.83 and 1.02 mg C3GE/L), respectively, followed by RE and PEF1 (1.26 and 1.35 mg C3GE/L). The lowest inhibition (*p* ˂ 0.05) was PEF2 (6.72 mg C3GE/L).

Dia et al. [70] found that soy compounds could also inhibit iNOS with a concentration of 1 mg/mL (66%). It was reported that extracts from hops at 1 and 10 μg/mL inhibited iNOS activity from 23% to 67%, respectively [71]. An extract from fresh blackcurrant berry (*Ribes nigrum* L.), which is an anthocyanin-rich source, was found effective for inhibiting COX-2 (IC_50_ = 1.77 ± 0.45 mg/L) and iNOS (IC_50_ = 0.61 ± 0.00 mg/L) [72]. In another work, 1 mg/mL of purple maize pericarp extract inhibited COX-2 (89.62%), and concentrations over 5 mg/mL effectively inhibited COX-1 [73]. These results show that natural ARE and fractions of them could inhibit inflammation-related enzymes.

Bioactive compounds in black bean could modulate molecular markers and affect the anti-inflammatory process in the tissues involved [74,75]. REF2 maintained its biological potential after the HPCCC process, showing a similar result compared to RE.

RE has high enzyme inhibition levels, as well as REF2 and PE. This inhibition could result from a synergistic effect between all the components in the extract, such as sugars, oligosaccharides, glycosylated derivatives of phenolics, and saponins, among others [22,31,60]. These three treatments were different: the first was the complete extract from black bean (RE), the second (PE) was the result before Amberlite^®^ resin, and REF2 was the most concentrated fraction obtained using HPCCC. They had different phytochemical profiles; nevertheless, they showed similar anti-inflammatory potential. Thus, the compounds interacting with the enzyme catalytic were retained regardless of the method applied [41]. The HPCCC process preserved almost all the compounds in the extract and retained the biological potential.

### 2.5. In Silico Analysis

Cyclooxygenase is an enzyme that catalyzes the synthesis of prostaglandins through the oxidation of arachidonic acid [75]. Prostaglandins carry out functions related to the homeostasis of various organs and pain, inflammation, and the development of neoplasia [76]. Both cyclooxygenases present a similar affinity for arachidonic acid and are 90% homologous. They have different affinities for the substrate and are found at different locations within the cell [77].

The iNOS is mainly expressed in macrophages exposed to bacterial, viral, or tumor affection and as a response to molecules such as bacterial lipopolysaccharides, cytokines, and necrosis factor tumor TNF-α [78,79]. iNOS is also expressed in other cell types, such as neutrophils, liver cells, cartilage chondrocytes, and muscle fibers [80].

In this study, the compounds identified in the studied common bean variety were evaluated as iNOS, COX-1, and COX-2 inhibitors by molecular docking. Table 4 shows that molecules found in the black bean extracts and fractions could interact with amino acid residues in the catalytic site of the enzymes and inhibit the production of pro-inflammatory molecules. For iNOS, the theoretical binding energies were better for ferulic acid (−6.7 kcal/mol), naringenin (−7.2 kcal/mol), and delphinidin-3-glucoside (−8.5 kcal/mol). These three phenolic compounds were present in black bean extracts and fractions, and their inhibition values are similar to L-arginine (−8.3 kcal/mol), a reported iNOS inhibitor. Therefore, these compounds could have been responsible for the anti-inflammatory potential determined in the in vitro inhibition assays.

Malvidin-3-glucoside (−8.45 kcal/mol), naringenin (−8.3 kcal/mol), and gallic acid (−7.2 kcal/mol) theoretical binding energy results were better for COX-1 interactions. A control inhibitor (mofezolac^®^) showed a value of −8.4 kcal/mol, indicating that these three compounds present a high potential to interact in the enzyme’s catalytic site. In the case of COX-2, better interactions were for catechin (−8.3 kcal/mol), ferulic acid (−8.2 kcal/mol), and cyanidin-3-glucoside (−7.5 kcal/mol). These molecules are phenolic derivates such as flavonoids, anthocyanins, and hydroxycinnamic acid, representing the diversity of compounds that could block this enzyme. It is known that in physiological conditions, COX-2 is inhibited by many compounds. A positive control inhibitor (celecoxib^®^) generated a −7.2 kcal/mol theoretical binding energy for this enzyme, which was less potent compared to some evaluated phenolic compounds.

Xu et al. [81] showed that an important group of diverse phenolic compounds in *Solanum xanthocarpum* presented good affinities (<−5 kJ/mol) with iNOS protein with affinities comparable to indomethacin.

According to Lescano et al. [82], a docking score ranging from −8.96 to −8.90 for COX-1 and scores ranging from −7.13 to −8.81 for COX-2 were obtained using polyphenols structures from *Campomanesia adamantium*. They also reported that depending on the polyphenol structure and polar regions, they could interact better with COX-1 or COX-2 because the proteins are promiscuous, influencing the change in affinity for COX-2 compared to COX-1. These results help to understand the molecular dynamics between protein-ligand attractions. Additionally, they show that the metabolites of interest were able to interact with the active sites of the three enzymes. The interaction depended on each component’s chemical structure and the enzyme’s active site [83].

Figure 4 shows the best conformation of the coupling reaction between iNOS and delphinidin-3-glucoside. iNOS and delphinidin-3-glucoside formed two hydrogen bonds (Figure 4a) between the sugar in the aglycone, tryptophan 372, and tryptophan 463 in the catalytic site with a theoretical binding energy of −8.5 kcal/mol. Glutamic acid 377, cysteine 200 and alanine 197 formed π bonds in the phenolic ring parts of the structure. Figure 4b shows the catalytic hydrophobic pocket in negative value and the conformations of the coupling of iNOS with delphinidin-3-glucoside.

Results presented here have shown that anthocyanin black bean molecules are good ligands for iNOS pro-inflammatory protein. This reactivity may explain the anti-inflammatory effects attributed to them. Moreover, anthocyanins have been reported to inhibit iNOS and COX-2 in cell-based assays [84].

Native Mexican black bean is an important source of natural pigments. Anthocyanins are reported in several works due to their high biological potential. This work reports the characterization and quantification of anthocyanin and phenolic compounds in extracts and fractions obtained from the “negro de vaina morada” black bean. Moreover, it was possible to produce anthocyanin-enriched fractions from native black beans using HPCCC with outstanding phytochemical profile and biological potential.

Moreover, other mixtures of solvents, pre-treatments, sources, and methodologies should be evaluated to improve the HPCCC performance. After optimization of the method, a high and diverse number of enriched-anthocyanin fractions could be produced. Fractions with outstanding phytochemical profiles and biological potential could be used for future studies. The study of native common bean cultivars represents an opportunity to evaluate their potential as a source of value-added ingredients for the food industry. Further experiments using biological models are needed to validate the antioxidant and anti-inflammatory potential of the black bean extracts and fractions. Characterizing and understanding the resin and solvents’ molecular interactions is important.

## 3. Materials and Methods

### 3.1. Materials

Black-seeded common beans (*Phaseolus vulgaris* L. cv Negro de vaina morada) were cultivated in November 2018 and collected in March 2019 in Chiapas, Mexico. Seeds were stored at 4 °C until use. Ultrapure water, ethanol (≥99%), sodium carbonate, 2 N Folin–Ciocalteu reagent, gallic acid, chlorohydric acid, potassium chloride, sodium acetate, Amberlite^®^ XAD-7HP, formic acid, n-hexane (≥95%), ethyl acetate (≥99.7%), methanol (≥99%), trifluoracetic acid (TFA), acetonitrile (≥99.9%), 2,2-diphenyl-1-picrylhydrazyl (DPPH), 2,2′-azino-bis(3-ethylbenzothiazoline)-6-sulfonic acid (ABTS), 6-hydroxy-2,5,7,8-tetramethylchrome-2-carboxylic acid (Trolox^®^), potassium persulfate, sodium nitroprusside, Griess reagent, gallic acid (≥99%), genistein (≥98%), myricetin (≥96%), apigenin-7-glucoside (≥90%), epigallocatechin gallate (≥80%), kaempferol-3-glucoside (≥95%), quercetin-3-glucoside (≥90%), quercetin-3-D-galactoside (≥98%), delphinidin 3-O-glucoside (≥98%), malvidin 3-O-glucoside (≥98%), and petunidin-3-O-glucoside (≥98%) were purchased from Sigma-Aldrich (St. Louis, MO, USA). iNOS inhibitor screening kit (Fluorometric) was purchased from Biovision (Milpitas, CA, USA). COX-1 (ovine) and COX-2 (human) inhibitor screening assay kits were purchased from Cayman Co. (St. Louis, MO, USA). CO_2_ gases were obtained from Grupo Infra (Guadalajara, Jal., México).

### 3.2. Supercritical Extraction

Extraction using supercritical fluids was performed using a Thar^®^ SFE500 Waters brand extractor (Thar Process, Pittsburgh, PA, USA), which was equipped with a 500 mL stainless steel extraction cell. The extraction conditions were 300 Bar, 60 °C, 10 g/min of a flow composed of 90% CO_2_ gas and 10% acidified ethanol/water 50/50 *v*/*v* (0.3% formic acid) as co-solvent, according to Hsieh-lo et al. [41]. In the extraction cell, 50 ± 1.0 g of whole “negro de vaina morada” black bean and 20 ± 1.0 g of boiling pearls were placed. Finally, the extract was concentrated on a rotary evaporator Buchi at 60 °C and 180 mbar. The concentrated raw extract (RE) was stored at −20 °C until analysis.

### 3.3. Purification by Column Chromatography

A total of 25 mL of RE was incorporated in the upper part of the packed Amberlite^®^ XAD-7HP column (30 × 3 cm). Then, acidified water (0.3% formic acid) was added in necessary volumes to remove impurities until the flow cleared. Subsequently, acidified ethanol/water 70/30 *v*/*v* (0.3% formic acid) was incorporated to elute the compounds until the flow came out crystalline. The extract was concentrated under rotary evaporation (Buchi) at 60 °C and 180 mbar. The concentrate purified extract (PE) was stored at −20 °C until analysis (Patent MX/a/2020/012916) [10].

### 3.4. Fractionation by High-Pressure Countercurrent Chromatography

The HPCCC (Dynamic extractions, London, UK) experiment was performed in analytical mode with a two-phase solvent system composed of n-hexane/ethyl acetate/methanol/water (2/2/3/2 *v*/*v*/*v*/*v*). First, the column was filled with the stationary phase (upper phase) with a flow rate of 5 mL/min, then the spectrum was rotated at 1600 rpm, and the mobile phase (lower phase) was pumped into the column at a flow rate of 1 mL/min. After reaching hydrodynamic equilibrium, 1 mL of sample (RE or PE) solution was injected into the column. The elution stage started with a flow rate of 1 mL/min until 35 min; then, extrusion started with a flow rate of 3 mL/min until 45 min. The effluent was monitored with UV detector set to the following: Channel A: 254 nm. Channel B: 280 nm C channel: 350 nm Channel D: 510 nm. The separation was carried out in reverse phase. Fractions were obtained considering UV detection. The characteristic wavelength used to detect anthocyanins is 520 nm. In Appendix A, the complete chromatogram for the method used is presented. REF1 and PEF1 correspond to the liquid flux obtained from 24 to 25.5 min of the run. REF2 and PEF2 correspond to the liquid flux obtained from 25.5 to 27 min of the run after the RE and PE were injected [53,85]. Appendix A shows the chromatogram obtained during the HPCCC experiment, where the fractions were collected along with the retention times associated with the anthocyanin’s absorbance. An experimental design diagram is shown in Figure 5. Four fractions were stored at −20 °C until analysis.

### 3.5. Determination of Total Phenolic Content and Total Anthocyanins Content

All determinations, procedures, and calculations were made in a liquid base due to the purification and fractionation processes. Total Phenolic Content was quantified using the Folin–Ciocalteu method reported by Mojica et al. [20]. Extracts or fractions were diluted with distilled water. In total, 50 μL of sample, distilled water (blank), or standard (gallic acid) were placed in a 96-well microplate. The calibration curve was made from 5–60 μg/mL of gallic acid. Subsequently, 50 μL of 1 N Folin–Ciocalteu reagent was added, mixed, and before adding 100 μL of 20% Na_2_CO_3_. Finally, 690 nm absorbance was read (Infinite M200 Pro, TECAN, Männedorf, Switzerland). Results were expressed in mg gallic acid equivalents per liter (GAE/L).

Total Anthocyanins Content was determined by pH differential method (AOAC 2005.02 official method). Extracts or fractions were diluted in two different buffers, 0.025 M KCl (pH 1.0) and 0.4 M CH_3_COONa (pH 4.5). Subsequently, aliquots of 200 μL of the diluted samples of each pH were placed in a 96-well microplate. The absorbance was read at 520 and 700 nm (Infinite M200 Pro, TECAN, Männedorf, Switzerland). Results were expressed in mg cyanidin-3-glucoside equivalents per liter (C3GE/L) [22].

### 3.6. Tentative Identification of Phenolic Compounds on Direct Analysis in Real-Time Mass Spectrometry (DART-MS)

DART-ESI-MS phenolic compound analysis was performed by the direct infusion on an ESI-QTof instrument Waters Xevo G2-XS QTof quadrupole time-of-flight mass spectrometry, equipped with an electrospray ionization (ESI) interface (Milford, MA, USA). MS acquisition process was performed on positive and negative ion modes. Set parameters were capillary voltage: 3.00 kV, cone voltage: 70 kV, temperature: 150 °C and desolvation temperature: 500 °C for positive ion mode and capillary voltage: 2.5 kV, cone voltage: 40 kV, and temperature: 100 °C and desolvation temperature: 250 °C for negative ion mode. Infusion flow rate was 5 µL/min. Mass range was from 50 to 800 *m*/*z*. Data acquisition and analysis were performed using the software MassLynx V4.1, Waters Corporation, Milford, MA, USA. Results were compared with the Mass Bank of North America (MoNA) to obtain the tentative mass prediction and identification [10,41,86,87].

### 3.7. Quantification of Anthocyanins and Phenolic Compounds by High-Pressure Liquid Chromatography

The methodology was carried out as reported by Alcázar-Valle et al. [8] with minor modifications. Extracts and fractions were diluted into ethanol (1% trifluoroacetic acid (TFA)) and refrigerated at 4 °C until analysis. PDA Detector (Acquity Arc, Waters, Milford, MA, USA) was used for HPLC analysis. Starting 90% A from 0–10, 82% A from 10–18 min, 72% from 18–19 min, 60% A from 19,239 min, and 90% A from 23–25 min were gradient conditions. Acidified water (0.1% TFA) was used as solvent A and acetonitrile (0.1% TFA) was used as solvent B. Flow rate was 0.7 mL/min. C18, 2.7 μm, 4.6 × 150 mm (Cortects, Waters, Milford, MA, USA) column was used. The oven was set at 30 °C, and 20 μL of the sample was injected. Phenolic compounds were quantified and analyzed at 280 nm, and Anthocyanin compounds at 520 nm. Standard curves were based on a range from 1 to 40 mg/L and prepared from pure standards. Results were expressed as mg/L of each phenolic compound or anthocyanin.

### 3.8. Antioxidant Potential

#### 3.8.1. DPPH Radical Scavenging

In total, 20 μL of sample (extracts or fractions), blank (distilled water), or Trolox^®^ standard (0.01–0.25 mM calibration curve) were placed in a 96-well microplate. Then, 180 μL of the DPPH radical (2.36 mg DPPH/100 mL ethanol) was added, mixed, and, before 30 min, a reading at 517 nm (Infinite M200 Pro, TECAN, Männedorf, Switzerland) was taken [88]. Results were expressed as IC_50_ values (mg C3GE/L).

#### 3.8.2. ABTS Radical Scavenging

In total, 3.5 mM ABTS and 1.225 mM Na_2_S_2_O_8_ in 100 mL of water were prepared by constantly stirring for 16 h in the dark until the radical was produced. Then, 20 μL of sample (extracts or fractions), blank (distilled water), or Trolox^®^ standard (0.01–0.25 mM calibration curve) were placed in a 96-well microplate. Subsequently, 180 μL of the ABTS radical was added, mixed, and 734 nm absorbance was read (Infinite M200 Pro, TECAN, Männedorf, Switzerland). Results were expressed as IC_50_ values (mg C3GE/L) [88].

#### 3.8.3. Nitric Oxide Species Inhibition

Nitric oxide assay was performed to evaluate the antioxidant activity. The Griess reaction was produced according to Giraldo [89] to determine the capacity of the extracts and fractions to capture nitric oxide (NO). Briefly, 50 µL of sample (extracts or fractions), control (water), or Trolox^®^ standard (0.01–0.25 mM calibration curve) were plated in a 96-well plate, and 50 μL 113 mM sodium nitroprusside was added. The plate was incubated at room temperature (120 min). Then, 100 µL of Griess reagent was added. The plate was incubated for 10 min at room temperature, and the absorbance was measured at 550 nm (Infinite M200 Pro, TECAN, Männedorf, Switzerland). Anti-nitrosative activity results were expressed as IC_50_ values (mg C3GE/L).

### 3.9. Anti-Inflammatory Potential

#### 3.9.1. Inducible Nitric Oxide Synthase Inhibition Assay

Inducible nitric oxide synthase (iNOS) is an enzyme that catalyzes nitric oxide (NO) production. Nitric oxide plays an important role in different biological processes related to inflammation. An iNOS inhibitor screening kit was used (Fluorometric) (Biovision, Milpitas, CA, USA) according to the manufacturer’s instructions. Inhibition of iNOS was measured by comparing the amount of NO produced in the presence of an inhibitor with the control background having no inhibitor in the samples [90]. Results were expressed as a percentage of inhibition of iNOS, and the calculation of IC_50_ (mg C3GE/L) was performed.

#### 3.9.2. Cyclooxygenases 1 and 2 Isoforms Inhibition Assays

Cyclooxygenase 1 and 2 (COX-1 and 2) inhibitor screening tests were conducted on each sample to evaluate their anti-inflammatory potential. The COX-2 (human) and COX-1 (ovine) inhibitor screening assay were performed using a kit (Cayman Co., St. Louis, MO, USA) according to the manufacturer’s instructions. This analysis is based on the production of Prostaglandin F2α by SnCl_2_ (Tin II chloride) reduction of COX-derived Prostaglandin H2 in the COX reaction. Both ovine COX-1 and human recombinant COX-2 enzymes were tested separately in this assay [91]. The results were expressed in the percentage of inhibition of COX-1 and -2, and IC_50_ was calculated (mg C3GE/L).

### 3.10. Molecular Docking

The theoretical interaction of identified phenolic compounds with the enzymes was evaluated by in silico analysis through molecular docking. Phenolic compounds 3D structures were downloaded from PubChem (www.pubchem.ncbi.nlm.nih.gov (accessed on 4 August 2021)). The crystallographic structures were obtained from Protein Data Bank (PDB; https://www.rcsb.org/ (accessed on 30 January 2023) for human iNOS, COX-1, and COX-2 with the PDB code and resolution 1M8E (2.90 Å), 3KK6 (2.75 Å), and 5IKR (2.34 Å), respectively. The 17 phenolic compounds and anthocyanin structures reported in Table 3 and mofezolac^®^, celecoxib^®^, and L-arginine (used as positive controls) were obtained from the PubChem database and were used as ligands. Molecules were prepared with Discovery Studio^®^ 4.0 to clean the enzyme structure from water molecules and ligands. Flexible torsions, charges, and grid size were assigned using Autodock Tools. Docking calculations were performed using Autodock Vina, and the binding pose with the lowest binding energy was selected to be visualized using Discovery Studio visualizer software 4.0 (Accelrys Inc., CA, USA) [91,92].

### 3.11. Statistical Analysis

The data obtained were analyzed using one-way ANOVA by StatPoint STATGRAPHICS Centurion XVI 16.1.03 statistical software (StatPoint Technologies, Inc., Warrenton, VA, USA). The assays were run in triplicate and performed in independent replicates. Statistical differences among independent variables were determined using Tukey’s Posthoc Test (*p* < 0.05). The IC_50_ values were calculated using GraphPad Prism software 8.0 (GraphPad Software, Inc., San Diego, CA, USA).

## 4. Conclusions

Countercurrent chromatography is an efficient alternative for separating and purifying natural bioactive compounds. The raw extract fraction 2 obtained by HPCCC presented the highest phenolic compounds and anthocyanins content and the highest antioxidant and anti-inflammatory potential. Native black bean anthocyanin-rich extracts and fractions evidenced the potential of these molecules to block inflammation-related enzymes, including iNOS, COX-1, and COX-2. The HPCCC technique preserves the compounds of interest and maintains the biological potential. HPCCC was more efficient in purifying compounds compared to the column purification process using resins. Even though this work evaluated the biological potential of extracts and fractions, in vitro and in vivo assays are needed to validate the native black bean anthocyanin-rich extracts biological potential. Native black bean anthocyanins could be used in the food industry as an ingredient for food formulation. Besides being natural pigments, these molecules could be used for their properties to modulate inflammation markers and neutralize free radicals, which could promote an improvement in human health. Native common bean varieties with unique phytochemical profiles could be used as a source of value-added ingredients for the food industry.

## Figures and Tables

**Figure 1 pharmaceuticals-16-00874-f001:**
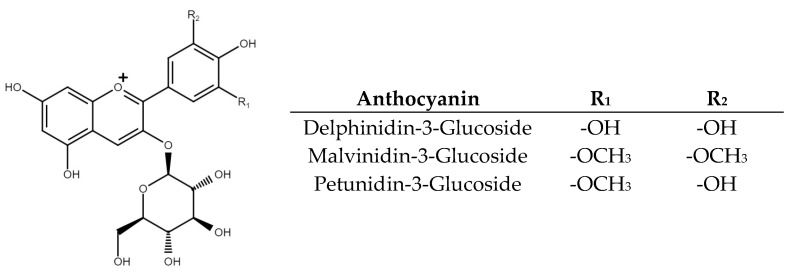
The chemical structure of the main anthocyanin in black beans. R_1_ and R_2_ represent the substituents in the first and second positions at the aromatic ring that change in each molecule.

**Figure 2 pharmaceuticals-16-00874-f002:**
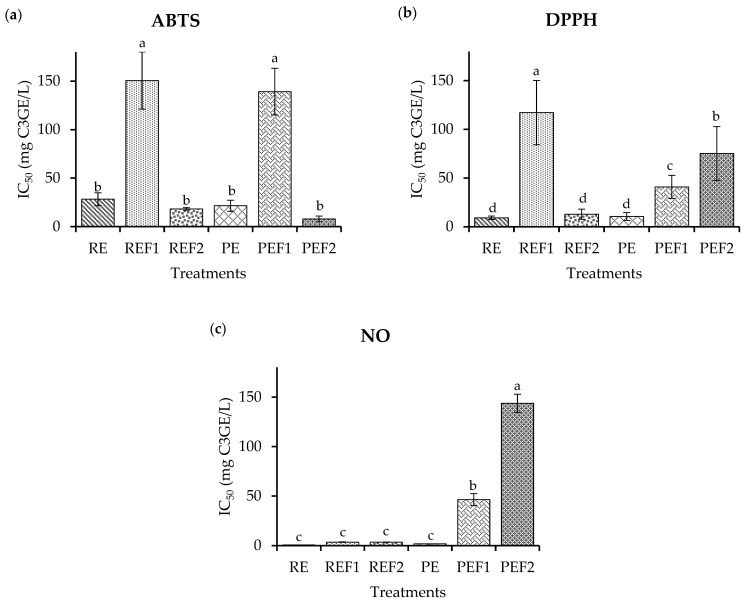
Antioxidant potential. (**a**) Radical ABTS inhibition essay. (**b**) Radical DPPH inhibition essay. (**c**) NO inhibition production essay. RE: Raw extract; REF1: Raw extract fraction 1; REF2: Raw extract fraction 2; PE: Purified extract; PEF1: Purified extract faction 1; PEF2: Purified extract fraction 2. Results are presented as the mean ± standard deviation with different lowercase letters denoting significant differences based on a post hoc Tukey’s test (*p* ˂ 0.05).

**Figure 3 pharmaceuticals-16-00874-f003:**
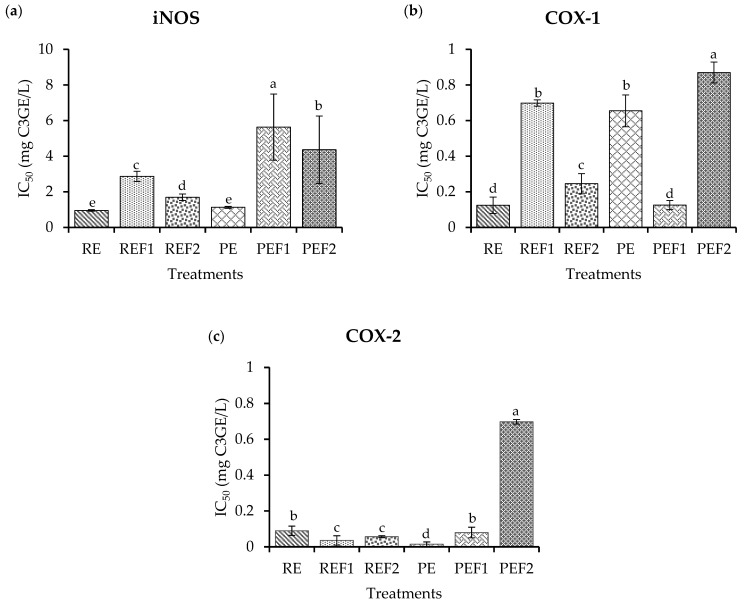
Anti-inflammatory potential. (**a**) iNOS inhibition essay. (**b**) COX-1 inhibition essay. (**c**) COX-2 inhibition essay. RE: Raw extract; REF1: Raw extract fraction 1; REF2: Raw extract fraction 2; PE: Purified extract; PEF1: Purified extract faction 1; PEF2: Purified extract fraction 2. Results are presented as the mean ± standard deviation with different lowercase letters denoting significant differences based on a post hoc Tukey’s test (*p* ˂ 0.05).

**Figure 4 pharmaceuticals-16-00874-f004:**
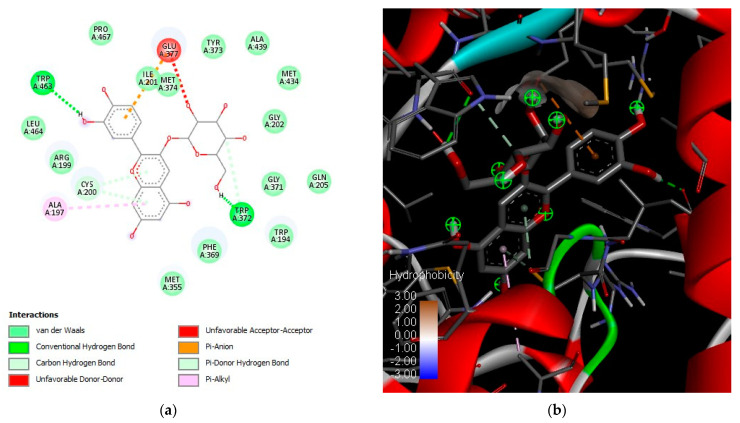
Molecular docking diagram of delphinidin-3-glucoside in the catalytic site of inducible nitric oxide synthase. (**a**) Amino acid residues’ main interactions with delphinidin-3-glucoside; (**b**) Best coupling conformation between iNOS catalytic site and delphinidin-3-glucoside.

**Figure 5 pharmaceuticals-16-00874-f005:**
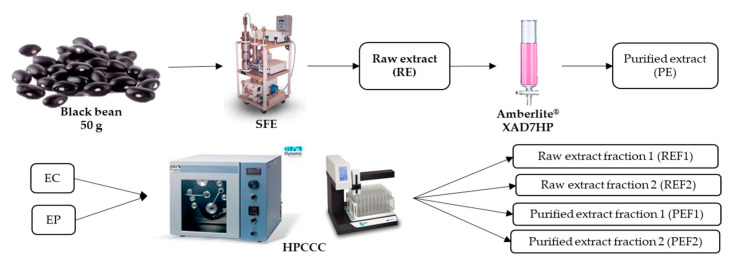
Experimental design diagram of treatments. SFE: Supercritical fluids extraction; RE: Raw extract; PE: Purified extract; HPCCC: High-performance countercurrent chromatography; REF1: Raw extract fraction 1; REF2: Raw extract fraction 2; PEF1: Purified extract fraction 1; PEF2: Purified extract fraction 2.

**Table 1 pharmaceuticals-16-00874-t001:** Total phenolic content and total anthocyanins in treatments.

Treatment	Total Phenolic Content (mg GAE/L)	Total Anthocyanin Content (mg C3GE/L)
RE	358.4 ± 24.3 ^a^	52.2 ± 4.4 ^b^
REF1	109.1 ± 8.2 ^c^	27.8 ± 0.1 ^c^
REF2	315.0 ± 15.8 ^a^	66.3 ± 1.1 ^a^
PE	185.4 ± 34.0 ^b^	28.1 ± 1.0 ^c^
PEF1	40.3 ± 10.8 ^d^	10.7 ± 0.5 ^d^
PEF2	48.0 ± 20.0 ^d^	13.1 ± 0.5 ^d^

RE: Raw extract; REF1: Raw extract fraction 1; REF2: Raw extract fraction 2; PE: Purified extract; PEF1: Purified extract faction 1; PEF2: Purified extract fraction 2; GAE: Gallic acid equivalents; C3GE: Cyanidin-3-glucoside equivalents. Results are presented as the mean ± standard deviation with different uppercase letters in the same column denoting significant differences based on a post hoc Tukey’s test (*p* < 0.05).

**Table 2 pharmaceuticals-16-00874-t002:** Phenolic compounds tentative identification by DART-MS spectrometry.

	RE	REF1	REF2	PE	PEF1	PEF2
Tentative Identification	*m*/*z* Experimental					
Quercetin-3-galactoside	463.1255	463.0462	463.0638	463.1255	463.055	463.0374
Malvidin-3-glucoside	331.0715	331.0752	331.0603	331.0715	331.0603	331.0603
Delphinidin 3-glucoside	303.0402	303.0794	-	303.0402	-	303.0794
Cyanidin 3-glucoside	447.1372	447.0766	447.0679	447.1372	447.0679	447.0766
Petunidin-3-glucoside	447.1285	447.0679	447.0766	447.1372	447.0679	447.0766
Gallic acid	169.0633	169.026	169.026	169.0686	169.0313	169.042
Genistein	269.189	269.2226	-	269.1957	269.2226	269.2293
Protocatechuic acid	153.07	153.0244	153.0244	153.07	153.0497	153.0497
Rutin	609.1727	-	-	609.1525	609.0716	-
Naringenin	271.0636	271.2356	-	271.1951	271.2423	271.2423
Catechin	289.0836	-	-	289.0975	289.0488	-
Glycitin	285.0322	285.1669	285.1635	285.0322	285.1635	285.1704
Myricetin	317.1151	-	-	317.1115	317.0677	317.075
Ferulic acid	193.0749	193.0493	193.0493	193.0749	193.0493	193.0493
p-coumaric acid	165.014	-	165.0508	165.0114	165.0486	-
Caffeic acid	179.0856	179.0829	179.0856	179.0911	179.0856	179.0637
Rosmarinic acid	359.1975	359.1432	-	359.1199	359.1509	359.1509

(-): No detection; RE: Raw extract; REF1: Raw extract fraction 1; REF2: Raw extract fraction 2; PE: Purified extract; PEF1: Purified extract faction 1; PEF2: Purified extract fraction 2.

**Table 3 pharmaceuticals-16-00874-t003:** Phenolic compounds and anthocyanin quantification by HPLC.

	Phenolic Compounds Concentration (mg/L)								Anthocyanins Concentration (mg/L)		
Treatment	GA	G	M	A7G	EGCG	K3G	Q3G	Q3Ga	D3G	M3G	P3G
RE	180.1	-	-	-	9.7	30.4	10.3	-	31.26	29.00	35.84
REF1	-	-	-	-	-	-	19.8	3.9	2.71	7.34	3.65
REF2	170.4	35.2	-	15.2	5.3	25.3	15.1	-	40.33	17.61	23.18
PE	-	-	10.4	-	14.8	39.5	70.4	15.2	4.51	5.91	3.17
PEF1	-	-	-	-	-	-	10.2	4.1	0.71	-	1.65
PEF2	-	-	-	-	-	30.2	14.6	-	3.40	16.81	5.90

(-): No detection; RE: Raw extract; REF1: Raw extract fraction 1; REF2: Raw extract fraction 2; PE: Purified extract; PEF1: Purified extract faction 1; PEF2: Purified extract fraction 2; GA; Gallic acid; G: Genistein; M: Myricetin; A7G: Apigenin-7-glucoside; EGCG: epigallocatechin gallate, K3G: Kaempferol-3-glucoside; Q3G: Quercetin-3-glucoside; Q3Ga: Quercetin-3-galactoside; D3G: Delphinidin-3-glucoside; M3G: Malvinidin-3-glucoside; P3G: Petunidin-3-glucoside.

**Table 4 pharmaceuticals-16-00874-t004:** Binding energy with inflammation process-related enzymes.

Compound	iNOS (kcal/mol)	COX-1 (kcal/mol)	COX-2 (kcal/mol)
Quercetin-3-galactoside	−1.8	−3.2	−3.3
Malvidin-3-glucoside	−5.3	−8.45	−6.3
Delphinidin 3-glucoside	−8.5	−4.3	−4.5
Cyanidin 3-glucoside	−3.4	−5.4	−7.5
Petunidin-3-glucoside	−4.2	−6.3	−6.3
Gallic acid	−2.5	−7.2	−2.8
Genistein	−1.6	−6.2	−1.8
Protocatechuic acid	−4.8	−1.4	−6.3
Rutin	−4.4	−5.6	−1.5
Naringenin	−7.2	−8.3	−4.3
Catechin	−6.4	−2.4	−8.3
Glycitin	−4.2	−3.5	−3.2
Myricetin	−3.4	−4.15	−4.2
Ferulic acid	−6.7	−2.36	−8.2
p-coumaric acid	−2.5	−6.2	−3.4
Caffeic acid	−3.8	−4.7	−4.4
Rosmarinic acid	−2.7	−5.2	−6.2
Mofezolac^®^	-	−8.4	-
Celecoxib^®^	-	-	−7.2
L-arginine	−8.3	-	-

(-): No evaluation. iNOS: inducible nitric oxide synthase; COX-1: ciclooxyganase-1; COX-2: cyclooxygenase 2.

## Data Availability

Data is contained within the article.

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
