# Peer review of "Mexican Native Black Bean Anthocyanin-Rich Extracts Modulate Biological Markers Associated with Inflammation"

_pharmaceuticals, 2023, doi:10.3390/ph16060874_

Round 1
Reviewer 1 Report
The novelty and the quality of the manuscript are good and it does not need extensive improvement before publication. It is carefully organized and written. It is easy to follow it and contains clear comments and conclusions. In my opinion, this manuscript is very detailed and meticulous, it covers all the literature in the field with critical point of view. The topic have been completely covered and is well connected through the text. There is a significant novelty in presented topic. For all these reasons, I can recommend the acception of the manuscript after minor revision:
1. I think that information about "Native common bean" could be extended, more examples should be added. This would be valuable for later publication citation.
2. The superiority of high-pressure countercurrent chromatography than other method of obtaining obtain ARE from different food matrices should be more emphasized.
3. The manuscript should be extended in scientific discussion. The authors presented their results and compared to some works, but did not present explanations for the reasons to reach these results.
4. Not all of the described results are covered in the discussion section.
5. No all information was given of the Identification of phenolic compounds by LC-ESI-QTof method. Is it is a newly developed or previously used method? Was it validated according to ICH guidelines?
Author Response
The novelty and the quality of the manuscript are good, and it does not need extensive improvement before publication. It is carefully organized and written. It is easy to follow it and contains clear comments and conclusions. In my opinion, this manuscript is very detailed and meticulous, it covers all the literature in the field with a critical point of view. The topic has been completely covered and is well connected through the text. There is a significant novelty in the presented topic. For all these reasons, I can recommend the acceptance of the manuscript after minor revision:
Author response:
Thank you for your comment.
- I think that information about "Native common bean" could be extended, more examples should be added. This would be valuable for later publication citation.
Author response:
A paragraph was added.
Pg 1, ln 36-42.
Recent studies have evaluated the biological potential, physicochemical and nutritional properties of native beans from the Phaseolus genera, including P. vulgaris, P. lunatus, P. polyanthus, and P. coccineous species. These beans were collected in Mexico's South Pacific Region, including Guerrero, Oaxaca, and Chiapas states. Local farmers select and preserve the best seeds for generations. This seed selection process could influence the phytochemical and nutritional content. Besides, those beans are a part of the culture and identity of the people in that region [8-10].
- The superiority of high-pressure countercurrent chromatography over other methods of obtaining obtain ARE from different food matrices should be more emphasized.
Author response:
Thank you for your comment, a paragraph was added to improve HPCCC differences.
Pg 2-3, ln 86-96.
HPCCC has been used to obtain anthocyanin-rich fractions from different food matrices. Winterhalter et al. [38-40] reported the extraction and fractionation of anthocyanins from natural sources such as Impatiens glandulifera, Vaccinium vitis-idaea L., black chokeberry, cranberry and pomegranate (Punica granatum L.) using HPCCC. The fractions obtained presented high anthocyanin concentration and a purified profile of the molecules of interest. HPCCC permits obtaining concentrated anthocyanin fractions with unique compositions from natural sources using less solvent and time [34,37,39,40]. However, few studies reported the biological potential of the generated fractions, physicochemical characteristics, and their phytochemical content. In this sense, HPCCC anthocyanin purified fractions could be characterized, and their biological potential and stability as colorants could be evaluated.
- The manuscript should be extended in scientific discussion. The authors presented their results and compared them to some works but did not present explanations for the reasons to reach these results.
Author response:
The discussion improved. Paragraphs were added.
Pg 4, ln 168-183:
Regarding the HPCCC method, the four solvents in the liquid mixture (water, hexane, ethyl acetate, and methanol) formed the upper and lower phases. The upper liquid phase was used as a stationary phase, and then the lower phase was used as a mobile phase through the column to separate the anthocyanin and phenolic compounds. The upper phase was a non-polar liquid mixture, while the lower phase was a polar liquid mixture. Phenolic compounds and anthocyanin are molecules with a polar structure, showing more affinity to the lower phase [12-13].
Accordingly, the first compounds eluted were those with high polarity present in the extracts (RE and PE), such as sugars and structures derived from these (0-24 min). Then, phenolic compounds and anthocyanins with intermediate polarity were eluted in a mixture (24-27 min) [12,16]. Other uncolored phenolic compounds, including flavonoids and phenolic acids present in the extracts, have a higher structural polarity compared to the anthocyanins and were collected in the first fractions of both extracts [34,39]. REF1 and PEF1 mainly contain uncolored phenolic compounds, while REF2 and PEF2 contain many colored components, principally glycosylated anthocyanins. At the end of the method (27-35 min), those components with a more non-polar structure were not collected.
Pg 7, ln 281-285:
Countercurrent chromatography is an efficient alternative for the separation and purification of natural bioactive compounds. The fraction REF2 obtained by HPCCC presented the highest phenolic compounds and anthocyanins content, similar to the RE. The efficiency of HPCCC was compared with the column purification process. HPCCC is a good methodology to isolate and purify bioactive compounds.
Pg 13, ln 471-479
Moreover, other mixtures of solvents, pre-treatments, sources, and methodologies should be evaluated to improve the HPCCC performance. After optimization of the method, a high and diverse number enriched-anthocyanin fractions could be produced. Fractions with outstanding phytochemical profiles and biological potential could be used for future studies. The study of native common bean cultivars represents an opportunity to evaluate their potential as a source of value-added ingredients for the food industry. Further experiments using biological models are needed to validate the antioxidant and anti-inflammatory potential of the black bean extracts and fractions. Characterizing and understanding the resin and solvents molecular interactions is important.
- Not all the described results are covered in the discussion section.
The discussion section was improved. Paragraphs were added as can be observed in the response to comment 3.
- No all information was given of the Identification of phenolic compounds by LC-ESI-QTof method. Is it a newly developed or previously used method? Was it validated according to ICH guidelines?
Author response:
Thank you for your comment. The method used is already published in recent articles [10,41,87]. Other groups (Gross, 2014) have reported this type of MS method (DART-MS) based on the direct injection of samples in the equipment without the pass for the chromatography column for the analysis. This technique permits the sample to reach the ionization section in the ESI. Finally, using the MoNA library, the methodology and mass were compared with available related methods and mass.
Reference
Gross JH. Direct analysis in real time--a critical review on DART-MS. Anal Bioanal Chem. 2014 Jan;406(1):63-80. doi: 10.1007/s00216-013-7316-0. Epub 2013 Sep 15. PMID: 24036523.
The methodology section was improved to read:
Pg 15 , ln 558-570
3.6. Tentative identification of phenolic compounds on direct analysis in real-time mass spectrometry (DART-MS)
DART-ESI-MS phenolic compound analysis was performed by the direct infusion on an ESI-QTof instrument Waters Xevo G2-XS QTof quadrupole time-of-flight mass spectrometry, equipped with an electrospray ionization (ESI) interface (Milford, MA, USA). MS acquisition process was performed on positive and negative ion modes. Set parameters were capillary voltage: 3.00 kV, cone voltage: 70 kV, temperature: 150 °C and desolvation temperature: 500 °C for positive ion mode and capillary voltage: 2.5 kV, cone voltage: 40 kV, temperature: 100 °C and desolvation temperature: 250 °C for negative ion mode. Infusion flow rate was 5 µL/min. Mass range was from 50 to 800 m/z. Data acquisition and analysis were performed using the software MassLynx V4.1, Waters Corporation, Milford, MA, USA. Results were compared with the Mass Bank of North America (MoNA) to obtain the tentative mass prediction and identification [10,41,86-87].
References added:
Virgen-Carrillo, C.A.; Valdés Miramontes, E.H.; Fonseca Hernández, D.; Luna-Vital, D.A.; Mojica, L. West Mexico Berries Modulate α-Amylase, α-Glucosidase and Pancreatic Lipase Using In Vitro and In Silico Approaches. Pharmaceuticals 2022, 15, 1081. https://doi.org/10.3390/ph15091081
Reviewer 2 Report
Interesting work on black beans anti-inflammatory capacity. I am inclined for it’s approval, with minor comments:
Introduction:
The paragraph about inflammation and chronic diseases fall a bit out of the rest; the text is well written and could be supported without this basic paragraph.
Also, I’d recommend a more ambitious goal description, as it is only a short sentence at the end of the introduction.
Results:
Some context is missing before the results are placed. Although the Material and Methods give all the details afterwards, the reader should be able to understand the study on it’s current format. I suggest a quick paragraph of “study description” either prior 2.1 or at the end of the introduction.
Apart from that, impressive results and writing.
Author Response
Interesting work on black beans anti-inflammatory capacity. I am inclined for its approval, with minor comments:
Author response:
Thank you for your comments.
Introduction:
The paragraph about inflammation and chronic diseases falls a bit out of the rest; the text is well written and could be supported without this basic paragraph.
Author response:
Thank you for your comment; the paragraph was modified to read:
Pg 2, ln 70-73:
Inflammation could increase oxidative stress in cells and tissues by producing reactive oxygen species (ROS) [25-26]. Black bean anthocyanin's anti-inflammatory and anti-diabetes biological potential have been reported [21-22,27].
Also, I’d recommend a more ambitious goal description, as it is only a short sentence at the end of the introduction.
Author response:
The objective of the manuscript was modified to read:
Pg 2, ln 97-99
This work aimed to obtain and characterize anthocyanin-rich extracts and fractions from Chiapas native black beans obtained using HPCCC and column purification and evaluate their antioxidant and anti-inflammatory potential using biochemical assays.
Results:
Some context is missing before the results are placed. Although the Material and Methods give all the details afterwards, the reader should be able to understand the study in its current format. I suggest a quick paragraph of “study description” either prior 2.1 or at the end of the introduction.
Author response:
Thank you for your comment, a paragraph clarifying the study was added:
Pg 3, ln 101-112:
2.1. Study description
A raw extract (RE) was obtained from native black beans by supercritical fluids technology. Then, RE was purified with Amberlite® XAD-7HP resin in a column (PE). RE and PE were injected into the HPCCC equipment to obtain anthocyanin-rich fractions. Two fractions from RE were collected accordingly with the anthocyanin retention time and absorption in the detector (REF1 and REF2). Similarly, another two fractions from PE were obtained (PEF1 and PEF2). Treatments (extracts (RE, PE) and fractions (REF1, REF2, PEF1, PEF2)) were evaluated to determine and identify the total phenolic and anthocyanin content. Some assays were evaluated in all treatments to probe the biological potential, including antioxidant and anti-inflammatory potential. Finally, an in-silico assay was performed to predict the theoretical interaction of the identified molecules of the common bean extract and fractions and molecular markers related to
Apart from that, impressive results and writing.
Author response:
Thank you for your comment.
Reviewer 3 Report
The paper is worth attention. It is connected to studies of beans. Supercritical extraction was used in the studies. The particular value I would give to the anti-inflammatory potential of the extracts and fractions ( by inhibiting enzymes related to the inflammation process, such as inducible nitric oxide synthase (iNOS), cyclooxygenase 1 (COX-1) and cyclooxygenase 2 (COX-2).
1. I recommend to provide the typical chemical structure of anthocyanins (line 44-52) for a better perception of the manuscript
2. It is necessary to provide information about fractions. Now it is understandable at all. This information is not present in the Results and discussion and Methods
3. 3.5. Justify why 690 nm was chosen for studies of TPC
4. Provide the aim of the studies of anti-inflammation activity and connect with the conclusion.
5. Conclusions. It is necessary to tie studies and conclusions. For instance, to relate to anti-inflammatory activity and so on. In conclusions there is accent on HPCCC technique ONLY
English is good, clear
Author Response
The paper is worth attention. It is connected to the study of beans. Supercritical extraction was used in the studies. The particular value I would give to the anti-inflammatory potential of the extracts and fractions (by inhibiting enzymes related to the inflammation process, such as inducible nitric oxide synthase (iNOS), cyclooxygenase 1 (COX-1) and cyclooxygenase 2 (COX-2).
Author response:
Thank you for your comment.
- I recommend providing the typical chemical structure of anthocyanins (line 44-52) for a better perception of the manuscript
Author response:
Thank you for your comment. A new figure and descriptions of it was added.
Pg 2, ln 58-67
Figure 1 shows the structure of the three main anthocyanins found and reported in black beans [10,20-22]. The figure evidence that the three molecules are different because they have different substituents, including methoxy and hydroxy groups in R1 and R2 positions in the phenolic ring.
Figure 1. The chemical structure of the main anthocyanins in black beans. R1 and R2 represent the substituents in the first and second positions at the aromatic ring change in each molecule.
- It is necessary to provide information about fractions. Now it is understandable at all. This information is not present in the Results and discussion and Methods.
Author response:
Thank you for your comment. A new part of a paragraph was edited.
Pg 14, ln 527-532
Fractions were obtained considering the UV detection. 520 nm is the characteristic wavelength used to detect anthocyanins. In supplemental Figure 1, the complete chromatogram for the method used is presented. REF1 and PEF1 correspond to the liquid flux obtained from 24 to 25.5 minutes of the run. REF2 and PEF2 correspond to the liquid flux obtained from 25.5 to 27 minutes of the run after the RE and PE were injected [53,85].
- 3.5. Justify why 690 nm was chosen for studies of TPC.
Author response:
There are many reports that use of Folin-Cicalteu reagent to determine the total phenolic content in different natural extracts. It is reported that the Folin reagent's maximum absorption level is between 650-730 nm. Other works reported measuring at 690 nm as a standard which is included within the range.
References
Blainski, A.; Lopes, G.C.; De Mello, J.C.P. Application and Analysis of the Folin Ciocalteu Method for the Determination of the Total Phenolic Content from Limonium Brasiliense L. Molecules 2013, 18, 6852-6865. https://doi.org/10.3390/molecules18066852
Mojica, L.; Meyer, A.; Berhow, M.; Gonzalez de Mejia, E. Bean cultivars (Phaseolus vulgaris L.) have similar high antioxidant capacity, in vitro inhibition of α-amylase and α-glucosidase while diverse phenolic composition and concentration. Food Res. Inter. 2015, 69, 38-48. https://doi.org/10.1016/j.foodres.2014.12.007
- Provide the aim of the studies of anti-inflammation activity and connect with the conclusion.
Author response:
Thank you for your comments, two new paragraphs were added.
Pg 7, ln 287-293
Antioxidant potential of treatments was related to the ability of the phenolic compounds and anthocyanins present in the extracts and fractions to neutralize oxygen free radicals that were generated in excess directly as the 2,2'-azino-bis(3-ethylbenzothiazoline-6-sulfonic acid (ABTS) and the 2,2-diphenyl-1-picrylhydrazyl (DPPH) or indirectly as the nitric oxide (NO) to the reaction environment. This antioxidant capacity could be related to activating the body's defense mechanisms that prevent the effects of oxidative stress [14,41,54].
Pg 9, ln 343-351
Inflammation is the biological process where cells respond to their environment when they detect a physicochemical change or mechanical stress. This condition, in general, is presented before other physiological pathologies such as obesity, hypertension, and type 2 diabetes [63]. It is known that molecules such as phenolic compounds present in natural products have the capacity to prevent and modulate inflammatory markers [64]. Some studies have reported that black beans anthocyanin could modulate molecular markers related to these diseases [22,65]. Nevertheless, there are no reports that evaluate how the biological potential is retained or modified after the extracts pass for purifying and concentrating process.
- Conclusions. It is necessary to tie studies and conclusions. For instance, to relate to anti-inflammatory activity and so on. In conclusions there is accent on HPCCC technique ONLY
Author response:
Conclusion was modified to read:
Pg 15-16, ln 650-665
Countercurrent chromatography is an efficient alternative for separating and purifying natural bioactive compounds. The raw extract fraction 2 obtained by HPCCC presented the highest phenolic compounds and anthocyanins content and the highest antioxidant and anti-inflammatory potential. Native black bean anthocyanin-rich extracts and fractions evidenced the potential of these molecules to block inflammation-related enzymes, including iNOS, COX-1, COX-2. HPCCC technique preserves the compounds of interest and maintains the biological potential. HPCCC was more efficient in purifying compounds compared to the column purification process using resins. Even though this work evaluated the biological potential of extracts and fractions, in vitro and in vivo assays are needed to validate the native black bean anthocyanin-rich extracts biological potential. Native black bean anthocyanins could be used in the food industry as an ingredient for food formulation. Besides being natural pigments, these molecules could be used for their properties to modulate inflammation markers and neutralize free radicals, which could promote an improvement in human health. Native common bean varieties with unique phytochemical profiles could be used as a source of value-added ingredients for the food industry.